# A Pilot Study on Ad Libitum Mediterranean Diet Intervention for Women with PCOS: A Mixed-Methods Exploration of Acceptability, Adherence, and Participant Lived Experience

**DOI:** 10.3390/nu17071105

**Published:** 2025-03-21

**Authors:** Nicole Scannell, Anthony Villani, Lisa Moran, Evangeline Mantzioris, Stephanie Cowan

**Affiliations:** 1School of Health, University of the Sunshine Coast, Sunshine Coast, QLD 4556, Australia; nicole.scannell@research.usc.edu.au (N.S.); avillani@usc.edu.au (A.V.); 2Monash Centre for Health Research and Implementation (MCHRI), School of Public Health and Preventive Medicine, Monash University, Melbourne, VIC 3168, Australia; stephanie.cowan@monash.edu; 3UniSA: Clinical & Health Sciences, Alliance for Research in Exercise, Nutrition, and Activity (ARENA), University of South Australia, Adelaide, SA 5000, Australia; evangeline.mantzioris@unisa.edu.au

**Keywords:** PCOS, Mediterranean diet, behaviour change wheel, COM-B, lived experience, dietary implementation

## Abstract

**Background/Objectives**: A healthy diet is essential for managing Polycystic Ovary Syndrome (PCOS), yet optimal recommendations remain unclear, highlighting the need to explore alternative lifestyle interventions. The Mediterranean diet (MedDiet) supports cardiometabolic health; however, challenges with adherence within this population are unknown. This study examines the acceptability and experiences of an ad libitum MedDiet in women with PCOS, offering recommendations for implementation. **Methods**: A 12-week MedDiet intervention was conducted with women aged 18–45 years, diagnosed with PCOS and a BMI ≥ 25 kg/m^2^ (*n* = 12). Adherence was assessed using the Mediterranean Diet Adherence Screener. Surveys and semi-structured interviews, guided by the Capability, Opportunity, Motivation–Behaviour (COM-B) model, explored participants’ experiences. Thematic analysis identified barriers and facilitators, which were mapped to the COM-B and Theoretical Domains Framework (TDF), with all findings subsequently aligned with the Behaviour Change Wheel to inform implementation strategies. **Results**: MedDiet adherence significantly improved from baseline to week 12 (Baseline: 4.1 ± 1.8; week 12: 8.3 ± 2.3; *p* = 0.001), alongside increases in knowledge (*p* = 0.004), cooking confidence (*p* = 0.01), and time management (*p* = 0.01). Adherence factors were mapped to 12 of the 14 TDF domains. Key facilitators included health benefits, reduced weight pressure, educational resources, and simple guidelines. Barriers involved organisation, food availability, and external influences. Effective implementation should integrate MedDiet education, behaviour change support, practical resources, and professional training for nutrition professionals and healthcare providers to support referrals and weight-neutral dietary management. **Conclusions**: A short-term ad libitum MedDiet is acceptable for women with PCOS. Strategies for patients and healthcare providers, aligned with the intervention functions of education, training, and enablement, are key to supporting adherence.

## 1. Introduction

Polycystic ovary syndrome (PCOS) is a complex and prevalent endocrine disorder that affects approximately 8–13% of women of reproductive age and is characterised by a spectrum of symptoms, including hyperandrogenism, ovulatory dysfunction, and polycystic ovarian morphology [1,2]. The syndrome is associated with hormonal and metabolic disturbances, such as intrinsic and extrinsic insulin resistance, and hyperandrogenism, which contributes to an increased risk of cardiovascular disease (CVD) [3,4], type 2 diabetes [5,6], and infertility [7]. Managing PCOS can be particularly challenging due to the multifactorial nature of the disorder, where metabolic, reproductive, and psychological symptoms intertwine [6,8]. For many women with PCOS, these complications are compounded by increased rates of depression and anxiety which adversely affect health-related quality of life and self-efficacy [9,10,11,12], contributing to barriers to maintaining effective lifestyle management strategies [13].

Lifestyle and weight management (defined as prevention of weight gain, achieving modest weight loss, and maintaining a reduced weight) remains a primary approach to PCOS management in the International Evidence-Based Guidelines for PCOS [2] and is associated with improved symptom profiles and metabolic outcomes [1,14]. As one component of this, weight loss for women with PCOS and overweight (BMI ≥ 25 kg/m^2^) or obesity (≥30 kg/m^2^), achieved through dietary intervention and physical activity, may improve insulin sensitivity, reduce hyperandrogenism, and enhance menstrual regularity [15,16]. Due to the absence of evidence for a preferred dietary approach, the most current international guidelines for PCOS management recommends adhering to a dietary pattern that is consistent with traditional healthy eating guidelines (e.g., dietary guidelines) [2,17,18].

However, women with PCOS frequently report difficulty adhering to dietary modification recommendations. Weight management goals also often feel elusive [19]. The barriers to weight management in PCOS could relate to a range of psychosocial and/or physiological factors including heightened body image distress, disordered eating behaviours, and frustration with dietary approaches that fail to address the unique physiological hurdles related to disrupted energy homeostasis in PCOS [19,20,21]. Women with PCOS report a strong need for individualised, weight-neutral support from healthcare professionals who understand the syndrome’s complexities [22,23]. However, weight bias within healthcare remains a significant barrier, deterring women from seeking support and worsening mental health outcomes [23,24]. At the same time, weight loss may be a personal health goal for some women with PCOS [23,25]. In such cases, weight-centred dietary strategies may be appropriate if implemented with professional guidance that considers both mental and physiological health and provides support and monitoring to ensure long-term well-being [26,27]. A systematic review found that women with PCOS have nearly three times the odds of experiencing an eating disorder compared to those without the condition [28], underscoring the need for cautious, individualised dietary support. Approaches that focus on improving dietary quality rather than weight loss may help reduce the risk of disordered eating and foster a healthier relationship with food [29]. Therefore, both weight-centred and weight-neutral dietary strategies are important in PCOS management. Sustainable dietary interventions should address the health risks of PCOS while considering the psychological and physiological challenges faced by this population. One such sustainable dietary approach may be the Mediterranean diet (MedDiet).

The traditional MedDiet originates from the olive-growing regions of the Mediterranean Basin and encompasses diverse culinary traditions [30]. Despite variations in its definition, the MedDiet is operationalised as a plant-based dietary pattern, emphasising a high intake of fruits, vegetables, whole grains, legumes, nuts, seeds, and extra-virgin olive oil (EVOO); moderate consumption of fermented dairy, eggs, poultry, and seafood; and minimal intake of red and processed meats, butter, vegetable oils, and ultra-processed foods [31]. The MedDiet has been extensively studied for its potential benefits in managing cardiometabolic consequences similar to those experienced in PCOS. Specifically, adherence has been inversely associated with central adiposity in epidemiological studies and with weight loss in dietary intervention trials, regardless of calorie restriction [32,33]. Additionally, it has been linked to improved insulin sensitivity, better glycaemic control, and reduced depressive symptoms, particularly in individuals with metabolic disturbances [34]. These findings have led to growing interest in the MedDiet as a potential dietary strategy for women with PCOS. Irrespectively, the global prevalence of and disease burden related to PCOS is increasing, including regions bordering the Mediterranean [35,36,37,38]. Although the prevalence and clinical manifestations of PCOS varies according to ethnicity and geographical living location [38], globalisation and urbanisation have led to important changes which may indeed favour the increasing burden of PCOS, including excessive caloric intake, obesity, and a movement away from traditional dietary patterns [39]. As such, there is evidence from Mediterranean populations, including both adult and younger populations, showing low to moderate adherence to a MedDiet in recent years [40,41,42,43]. Nevertheless, the proposed biological mechanisms underpinning the potential therapeutic benefits of a MedDiet for the management of PCOS features have previously been reported [44]. Moreover, in a sample of *n* = 94 Italian women with PCOS (BMI: 38.2 ± 6.6 kg/m^2^; 24.1 ± 3.6 years), Barrea et al. [45] reported that low MedDiet adherence was predictive of a ‘metabolically unhealthy’ phenotype of PCOS. Nevertheless, despite the proposed benefits [44], adherence to a MedDiet can be challenging, particularly in non-Mediterranean countries. A cross-sectional study of Australian adults (*n* = 606) identified key barriers, including limited knowledge, high costs, and time constraints [46]. Other studies suggest that while adherence is possible, tailored support may be required to overcome specific barriers [47,48]. Although the benefits of a MedDiet are well recognised, research gaps persist regarding its acceptability for women with PCOS [49,50].

The Behaviour Change Wheel (BCW) provides a comprehensive framework for designing and evaluating interventions to support dietary adherence [51]. Central to the BCW is the Capability, Opportunity, Motivation–Behaviour (COM-B) model, which conceptualises behaviour as a function of these three interacting components [52]. The COM-B model has been widely applied to identify barriers and facilitators to lifestyle modifications, including dietary and physical activity changes in women with PCOS [53,54,55]. To further refine behavioural analysis, the Theoretical Domains Framework (TDF) extends the COM-B model by categorising 14 psychological, social, and environmental determinants of behaviour [56]. The BCW guides researchers in selecting effective intervention functions (broad behavioural change strategies such as education, persuasion, and environmental restructuring) as well as relevant policy categories that address structural, economic, or social factors influencing behaviour. Finally, the behaviour change taxonomy outlines 93 Behaviour Change Techniques (BCTs) that can be employed to deliver specific, evidence-based strategies required to implement these intervention functions and policy categories, ensuring that interventions are contextually relevant and sustainable [57].

While general barriers to MedDiet adherence have been documented [46], little is known about its acceptability and perceived challenges among women with PCOS. To address this gap, this mixed-methods study applied the BCW to explore the lived experiences of women with PCOS and a BMI ≥ 25 kg/m^2^, who were randomised to a MedDiet intervention in a 12-week pilot study comparing an ad libitum MedDiet intervention to a control. Findings provide a comprehensive understanding of the factors influencing MedDiet adherence and are used to develop recommendations to support long-term adherence and improved health outcomes in PCOS.

## 2. Materials and Methods

### 2.1. Study Design

This study used a concurrent mixed-methods design, where quantitative and qualitative data were collected and analysed in parallel. This approach can strengthen evidence through confirmation and substantiation of findings and can help to offset inherent methodological biases or weaknesses associated with the two different forms of inquiry [58]. Additionally, an advisory committee consisting of an obstetrician, endocrinologist, dietitian, and consumer representative provided strategic guidance on the study protocol and the development of educational resources to enhance engagement with the target population. The efficacy of the MedDiet intervention (e.g., changes in hormonal, metabolic, and anthropometric outcomes) is not reported here and will be reported in a future publication. In the present study, we report on intervention adherence, acceptability, and the lived experience of study participants following the MedDiet intervention.

Participants completed a 12-week randomised controlled trial (RCT) designed to investigate the efficacy and acceptability of a MedDiet intervention, without calorie restriction in women with PCOS and a BMI of ≥25 kg/m^2^. A comprehensive study protocol has been published elsewhere [59]. Briefly, participants were randomly allocated to receive either an ad libitum MedDiet intervention or a Healthy Eating control. Independent of their allocation (intervention or control), all participants received fortnightly dietary consultations from an Accredited Practicing Dietitian (APD), dietary and health education, weekly digital messages, meal suggestions and recipes, and other educational resources. The intervention framework [59] was developed using the COM-B model in combination with elements of the behaviour change technique taxonomy [60] to ensure the intervention components effectively addressed the behavioural drivers of dietary change.

Anthropometric (weight, BMI, and waist circumference) and biochemical (total testosterone, sex-hormone-binding globulin, fasting insulin, fasting glucose) measures, physical activity levels (International Physical Activity Questionnaire-Short form), and the Mediterranean Diet Adherence Screener (MEDAS) were collected at baseline and week 12. Four-day food records were completed at baseline, week 6, and week 12, and food checklists were collected fortnightly throughout the 12-week intervention. In addition, participants allocated to the MedDiet intervention also completed survey questions and participated in semi-structured interviews at baseline and week 12 to assess intervention acceptability. This multi-centre RCT was conducted from July 2021 to September 2023, with participating sites at the University of the Sunshine Coast (Queensland, Australia), Monash University (Victoria, Australia), and the University of South Australia (South Australia, Australia). This trial has been registered with the Australian and New Zealand Clinical Trials Registry (ACTRN12621000994886).

#### 2.1.1. Inclusion Criteria

Inclusion criteria included written confirmation of a PCOS diagnosis from a medical doctor, a BMI of ≥25 kg/m^2^, aged between 18 and 45 years, and not currently pregnant. Exclusion criteria included the use of insulin-sensitising medications or hormonal contraceptives within 3 months of commencement of the trial, as well as existing medical conditions, including Cushing syndrome, diabetes types 1 or 2, thyroid conditions, active cancer, or adrenal tumours. Individuals with a high adherence to a MedDiet (MEDAS score ≥ 10) were also excluded.

#### 2.1.2. Mediterranean Diet Protocol

An ad libitum MedDiet protocol was provided to participants, detailing the recommended daily and weekly quantities of food groups consistent with a traditional MedDiet eating pattern, as presented in Table 1.

#### 2.1.3. Control Group

The control group followed a dietary protocol consistent with general population-based dietary recommendations in accordance with the Australian Dietary Guidelines and Australian Guide to Healthy Eating [61]. Participants randomised to this group received the same number and type of resources, including dietary consultations and text messages [59]; however, this paper will focus on the lived experiences of following the MedDiet only.

#### 2.1.4. Dietary Resources

Following randomisation, participants received a resource pack containing a range of educational resources, including (1) a fridge magnet depicting key dietary guidelines and health messages; (2) an information pamphlet explaining the health effects of PCOS and benefits of dietary management; (3) a folder containing a visual representation of food categories and serving sizes, food checklists for daily diet recording, and meal suggestions with recipes. Additional resources providing meal suggestions and recipes were provided during each dietary consultation and were tailored to accommodate individual taste and cultural preferences. Given that the MedDiet is based on key dietary principles as opposed to a rigid and prescriptive protocol, it can be individualised to fit different cultural and culinary traditions. As such, only minor adjustments are needed to align traditional cuisines with principles of the MedDiet [62]. For example, a variation from traditional Indian cuisine which may include a tomato-based or dry vegetable curry in replace of a coconut cream base curry, cooked with olive oil in substitute of vegetable oil or ghee, and serving it with brown rice rather than white rice or naan bread.

#### 2.1.5. Dietary Consultations

At the baseline appointment, participants randomised to receive the MedDiet intervention received one-on-one counselling and education on MedDiet principles using the aforementioned dietary resources. Participants then attended six fortnightly dietary consults in-person or via Zoom (dependant on participant preference and living location). Each lasted 30 min and included dietary education and counselling and personalised goal setting.

#### 2.1.6. Digital Messaging

Weekly health prompts were sent to participants’ mobile phones via text messages. These one-way messages addressed key components of the intervention and highlighted helpful strategies for dietary adherence, such as “*Did you know legumes stabilise blood sugar levels while keeping us full? Need inspiration? Beans on toast, add lentils to a salad or snack on roasted chickpeas.” and “Having support from someone close to you can help you reach your goals. Spend time with someone who will encourage you to maintain a Mediterranean eating style*.”

### 2.2. Data Collection

#### 2.2.1. Participant Demographics

Participants completed a series of sociodemographic questions covering country of birth, cultural background, age, smoking status, education, socioeconomic status, and previous medical conditions.

#### 2.2.2. Adherence to a MedDiet Intervention

Adherence toward the MedDiet intervention was assessed at baseline and week 12 using the validated 14-item MEDAS [63]. This tool assesses adherence based on the habitual frequency of consumption of 12 main dietary components and two food habits consistent with a traditional MedDiet pattern. Items were scored dichotomously as either 0 or 1, with the total score reflecting adherence levels. A MEDAS score of ≥10 suggests high adherence, scores of 6 to 9 indicate moderate adherence, and scores of ≤5 indicate low adherence.

#### 2.2.3. Acceptability to a MedDiet Intervention

Acceptability toward the MedDiet intervention was assessed through a combination of survey questions and semi-structured interviews (Appendix A). Development of the survey tools and interview schedule were informed by the COM-B theoretical model proposed by Michie et al. [64]. Questionnaires were reviewed, discussed, and piloted by the stakeholder committee, which included representative PCOS experts in the form of clinicians, researchers, and a consumer representative, to provide face validity [65].

#### 2.2.4. Surveys

At baseline and week 12, participants completed a written survey designed to assess their confidence and perceived ability to overcome common barriers to adherence, including food access, nutrition knowledge, cooking skills, and motivation to change. The survey consisted of eight statements, each evaluated using a 5-point Likert scale, where 1 indicated “very low” or “no confidence/ability”, and 5 indicated “very high” confidence/ability. Additionally, at week 12, participants completed a written survey aimed at evaluating the delivery of the intervention. This survey included nine statements assessing the usefulness and ease of use of the educational sessions, resources, and text messages provided during the intervention. Responses were rated on a 5-point Likert scale, with 1 indicating “strongly agree” and 5 indicating “strongly disagree”.

#### 2.2.5. Interviews

Upon completion of the intervention (week 12), women participated in a semi-structured interview conducted in person or via online video conferencing (Zoom). The interviews were conducted by an Accredited Practising Dietitian (NS) with qualitative research experience and training. NS was the study dietitian, who worked closely with study participants, forming constructive relationships that enabled questioning on more sensitive topics. Immediate post-interview memoing was conducted to capture contextual details and noteworthy nuances, ensuring a richer interpretation of the data. The interviews lasted approximately 25 min, and the question schedule consisted of 10 open-ended questions that addressed participants’ barriers and enablers to following the MedDiet intervention, covering all domains of the COM-B to understand the factors affecting dietary adherence, including capability (e.g., what skills are required), opportunity (e.g., educational resources provided), and motivation (e.g., perceived health and lifestyle impacts and feelings towards following an ad libitum dietary protocol without structured weight loss).

### 2.3. Statistical Analysis

The primary aim of the pilot RCT was to assess changes in HOMA-IR. Accordingly, the target sample size was set at *n* = 42 participants (*n* = 21 MedDiet; *n* = 21 Control), based on 80% power to detect a significant (*p* < 0.05, two-sided) change in HOMA-IR of 1.7 ± 0.5, assuming a 30% attrition rate. However, actual recruitment numbers were substantially lower, as outlined in Section 3 and will be further reported on in a future feasibility publication. This has also been outlined elsewhere [59]. Moreover, the concept of information power was retrospectively considered for the sample size calculation of qualitative data. A moderate sample of 10–15 participants is appropriate, given the narrow aim (barriers and enablers to following a MedDiet in women with PCOS and BMI ≤ 25 kg/m^2^), purposive sampling of participants specific to the research aim (women with PCOS), use of theory (BCW, COM-B, TDF) to underpin data collection, and clear communication between researchers and participants [66].

#### 2.3.1. Quantitative Data

Descriptive statistics were used for participant demographics and survey responses related to intervention delivery with continuous variables presented as the mean (±SD) and categorical variables presented as frequencies or percentages. Paired t-test was used to assess group changes in MedDiet adherence scores from baseline to week 12. A Wilcoxon signed-ranks test was used to identify changes in survey responses and presented as the median. All quantitative analyses were performed using Statistical Package for the Social Sciences (SPSS) for Windows 26.0 software (IBM Corp., Armonk, NY, USA), with statistical significance set at *p* ≤ 0.05.

#### 2.3.2. Qualitative Data

All interviews were audio-recorded, and any noteworthy contextual details (e.g., non-verbal cues and emotional responses) and insights (e.g., novel perspectives or conflicting information) identified by the researcher were annotated in the participants record. Interviews were transcribed verbatim by researcher NS. Framework analysis [67] was conducted using NVIVO software (QSR International, Melbourne, Australia), version 12. The framework method provides clear steps to follow and produces highly structured outputs of summarised data. Data analysis involved familiarisation with the data and open coding, where codes were iteratively reviewed and clustered into conceptually related categories to develop a working analytical framework, which was then applied to the entire dataset and charted into a framework matrix, allowing for greater interpretation of the data and theme generation [67]. Memos were employed throughout the analysis process to facilitate the organisation and understanding of the data and guide coding and categorization. Analysis was led by one researcher (NS), in constant discussion and consensus with AV, who independently completed 30%. Any discrepancies were discussed and unanimously decided upon by all authors.

#### 2.3.3. Data Integration

Integration can occur at multiple stages of the research process; however, in this study, it was applied at the interpretation stage. At this level, data merging was conducted following the thematic analysis of the qualitative data and statistical analysis of the quantitative data [68,69]. Data transformation was not necessary, as both data types could be directly compared and interpreted in their original formats. Joint displays were used to synthesise the two data types to generate a more comprehensive understanding of the research findings.

Themes generated from the qualitative interviews and findings from the quantitative surveys were then mapped to the COM-B and TDF constructs to explore behavioural determinants of dietary adherence guided by the Behavioural Change Wheel [64]. To facilitate future intervention refinement, these COM-B components were matched to the relevant intervention functions and policy categories, as guided by the BCW. Targeted interventions were then developed by applying the APEASE (acceptability, practicability, effectiveness and cost-effectiveness, side effects, and safety and equity) framework and by identifying behaviour change techniques (using BCT taxonomy version 1) to deliver practical and specific recommendations.

## 3. Results

### 3.1. Recruitment

The results presented in this paper focus solely on the MedDiet group, as this is the primary focus of this study.

An overview of participant recruitment is shown in Figure 1. Initial interest in study participation was *n* = 380. A total of *n* = 40 participants met the inclusion criteria and were eligible to participate. Of the 34 who agreed to participate, 8 were unable to attend due to competing priorities or unexplained reasons. Among the 26 participants who were randomised, 12 commenced the MedDiet intervention, with 10 successfully completing the study.

### 3.2. Participants

Baseline characteristics of the participants are summarised in Table 2. A minority of participants (17%) reported taking additional medications for medical conditions, including reflux, asthma, and anxiety/depression. Supplement use was reported in one-third of women and included collagen powder, inositol, N-acetyl cysteine, probiotics, vitamin D, and mixed formulations, such as those for hormone balance and liver detox. The majority of women were Australian (75%), with 83% reporting a household income of less than $125,000 and over half (67%) holding a university degree.

### 3.3. MedDiet Adherence

Adherence to a MedDiet increased significantly from baseline to week 12 for participants randomised to receive a MedDiet intervention (*n* = 10) (baseline: 4.1 ± 1.8; week 12: 8.3 ± 2.3; *p =* 0.001) (Figure 2).

### 3.4. Surveys

#### 3.4.1. Behavioural Analysis

Self-reported ratings showed that participants felt their knowledge of a MedDiet significantly increased from baseline to week 12 (Z = −2.88, *p* = 0.004), as did the participants confidence to prepare/cook the included foods (Z = −2.55, *p* = 0.01) and confidence to have time needed to cook/prepare meals (Z = −2.59, *p* = 0.01) (Figure 3). Their ability to afford (Z = −2.45, *p* = 0.01) and access foods (Z = −2.11, *p* = 0.035) also significantly increased after the intervention. There was no significant change in their intention to follow a MedDiet, ability to adhere to a MedDiet, and acceptability of a MedDiet by friends and family.

#### 3.4.2. Intervention Delivery

Participants assessed the study resources and delivery of the dietary intervention, as presented in Figure 4. All participants strongly agreed that attending the dietary consults was useful (100%). The study resources were assessed as being easy to read, easy to understand, and useful by all participants (100%), with most participants either strongly agreeing or agreeing that they made dietary adherence easier (90%). Most participants agreed or strongly agreed that the one-way weekly text messages were helpful (80%) and not too short in length (100%). However, assessment of the number of messages was mixed, with half the participants (50%) wanting more messages and around one-third of participants (30%) neither agreeing nor disagreeing to increasing the number of messages.

### 3.5. Interviews

Thematic analysis revealed a total of nineteen themes, encompassing five themes mapped to Capability, four themes to Opportunity, and ten themes to Motivation. These themes were mapped to 12 TDF domains and are presented with representative quotes in Table 3.

COM-B: Psychological Capability—TDF Knowledge, Memory, Attention, Decision Processes, and Behaviour Regulation

Education and resources

Participants emphasised the value of structured education provided through dietary consultations, which helped to clarify what constitutes a MedDiet and how to practically implement it. Resource materials to support education sessions, reiterating the key learning points from the session and including practical applications such as recipes, were specifically noted as helpful in supporting adherence. Participants also reported that learning about the benefits of the MedDiet, understanding the nutritional value of various food groups, and debunking food myths contributed to their ability to follow the diet consistently.

Simplicity of guidelines

Participants described the Mediterranean diet as straightforward and easy to integrate into their daily routines. The flexibility of following broad food group guidelines, rather than strict nutrient targets, reduced the cognitive burden associated with dietary change.

Habitual diet

Long-standing dietary habits, particularly the frequent consumption of red meat or routine take-away meals, posed challenges for some participants in aligning their eating patterns with MedDiet recommendations.

Organisation and planning

Participants acknowledged that planning meals, snacks, and grocery shopping in advance facilitated dietary modification. However, many found this challenging, as it was not part of their usual routine. When planning did not occur, participants reported difficulties in meeting recommended food quantities and types, as well as challenges in making nutritious food choices.

COM-B: Physical Capability—TDF Skills

Culinary skills

Participants noted that cooking skills were particularly useful in improving variety, palatability, and overall enjoyment of the diet. Participants with limited culinary experience found it more challenging to prepare diverse, flavourful meals that aligned with MedDiet principles. Over time, participants reported that their cooking skills improved, often facilitated by strategies such as using pre-cut vegetables or opting for frozen and tinned foods (e.g., vegetables and fish). These approaches helped reduce preparation time and complexity, making adherence more manageable.

COM-B: Social Opportunity—TDF Social Influence

External influences

Supportive behaviours from friends and family, such as sharing MedDiet-compatible meals, provided positive reinforcement and encouragement to stay committed to the diet facilitated dietary adherence. However, misaligned food choices from family members, colleagues, or social groups made adherence more difficult, particularly during shared meals or social gatherings.

COM-B: Physical Opportunity—TDF Environmental Context and Resources

Time

Some participants found the MedDiet easy to follow and did not feel it required additional time, while others considered it time-intensive, especially due to the focus on preparing fresh foods and meals.

Cost

Participants who reduced their spending on dining out or alcohol found the MedDiet to be cost-effective. Others considered its cost to be comparable to that of their usual diet. However, some participants were concerned about the expense of core MedDiet foods, such as fresh seafood, olive oil, and nuts, and identified this as a barrier.

Availability

Participants appreciated the ease of being able to obtain fruits, vegetables, and other core MedDiet foods at regular supermarkets. However, some participants reported difficulty sourcing certain items, such as fresh seafood. Additionally, the limited availability of MedDiet-friendly options at restaurants and fast-food establishments posed barriers, making it more challenging to adhere to the diet when eating out.

COM-B: Reflective Motivation—TDF Belief About Capabilities, Optimism, Intention, Belief About Consequences

Self-discipline

Will power and restraint were identified by participants as key to resist temptations and consistently make choices that were aligned with the MedDiet.

Improved self-efficacy

Participants reported increased confidence and motivation to make healthier lifestyle choices. This extended beyond diet to other areas, such as physical activity, reflecting a broader shift in health behaviours.

Improved food relationship

Participants appreciated the flexibility to choose foods based on personal preference, rather than being restricted due to calorie counting. This shift was supported by dietary guidelines emphasising inclusion rather than exclusion, which participants found liberating and sustainable.

Willingness to change/adapt

Participants expressed an openness to modifying their habitual dietary patterns, trying new foods, and troubleshooting challenges. This adaptability made it easier to overcome barriers and sustain adherence.

Intention to continue

Participants perceived the diet as a long-term, sustainable choice rather than a temporary intervention, which both reinforced and facilitated adherence.

Perceived health benefits

Participants listed a range of health improvements they associated with the MedDiet, including enhanced fertility, improved insulin resistance, better mood, increased longevity, and improved diet quality. These perceived benefits were frequently cited as motivators which facilitated adherence.

COM-B: Automatic Motivation—TDF Reinforcement, Emotion

Improved health

Participants reported experiencing numerous benefits, including increased energy, better sleep, reduced stress, clearer skin, and improved anthropometric measures (e.g., weight and waist circumference).

Stress

Participants noted that periods of heightened stress, often related to work, study, or personal circumstances, reduced their motivation and capacity to adhere to the MedDiet.

Reduced weight pressure

Participants valued the diet’s weight-neutral approach, which removed the stress of weight monitoring and allowed them to focus on overall health improvements.

Enjoyable experience

The intervention was characterised as positive, fun, and adventurous, with participants noting that this enjoyment motivated them to maintain adherence.

### 3.6. Behavioural Change Strategies from the Intergration of Quantitative and Qualitative Data

Integration of qualitative and quantitative data reveals a strong alignment in the factors influencing capability and opportunity to adhere to the MedDiet (Figure 5). Both data sources consistently identified behavioural strategies that enhance food access and affordability, time management, cooking skills, MedDiet knowledge, and social support as critical to improving dietary adherence. This strong alignment underscores the importance of addressing these common barriers to dietary implementation when designing MedDiet interventions. Additionally, the qualitative data suggest that effective MedDiet interventions should incorporate behavioural strategies that frame the diet as flexible, focused on nutrient inclusion rather than emphasising restriction, and being weight-neutral.

Using the Behaviour Change Wheel framework, recommendations for intervention strategies encompassed six intervention functions: education, training, enablement, persuasion, incentivisation, and modelling, facilitated by three policy categories: communication/marketing, service provision, and guidelines. To enhance Capability, education, training, and enablement are essential for both women with PCOS and health professionals. Women require targeted education on applying principles of the MedDiet to PCOS, myth-busting, implementation strategies, practical resources, and cooking demonstrations. Ensuring health professionals have access to training on the current evidence base and are equipped with practical tools to deliver client-centred strategies should also be considered. These strategies should be primarily delivered through communication/marketing and service provision. Opportunity can be improved through enablement, education, training, and modelling, helping patients overcome practical barriers, navigate social influences, and build supportive networks. For health professionals, persuasion and incentivisation can be used to promote referrals to nutrition professionals by highlighting their ability to support patient self-efficacy and long-term dietary adherence while aligning with government-funded payment schemes. These can be facilitated through communication/marketing, service provision, and guidelines. Lastly, to enhance motivation, education, persuasion, and enablement can be used to reframe diet and lifestyle behaviours as positive, self-care-focused, and joyful experiences that provide broad health benefits. Additionally, health professionals may benefit from education on weight-neutral approaches that focus on nutrition inclusion for PCOS management. These strategies should be implemented through guidelines, service provision, and communication/marketing. The recommended strategies for dietary implementation are presented in Figure 5. Presented in Appendix A is a comprehensive description of these strategies aligned with the relevant intervention functions, policy categories, and behavioural change techniques.

## 4. Discussion

This study demonstrates that a MedDiet intervention in women with PCOS significantly enhanced participants’ knowledge of the MedDiet, as well as their ability and confidence to adhere to this eating pattern. Furthermore, adherence to the MedDiet was acceptable, as evidenced by a significant increase in MEDAS scores post-intervention. Participants identified a range of barriers, including time, cost, availability, and external influences and facilitators, such as perceived health benefits, reduced weight pressure, education and resources, and simplicity of dietary recommendations. These findings suggest that the MedDiet may represent a viable and sustainable dietary strategy for women with PCOS, particularly when intervention components are designed to address both practical and motivational barriers.

Consistent with previous research, our data indicate that adherence to the MedDiet is impeded by several barriers that are also observed across healthy eating interventions [70,71], including time constraints, cost, and food availability. In the present study, time emerged as a significant barrier, primarily due to the reliance on less convenient food options, a challenge similarly reported in a European study where time constraints were also identified as a barrier to healthy eating and were associated with a reduction in vegetable intake [71]. Although fresh fruit and vegetables were generally accessible, participants noted difficulties in sourcing fresh fish and seafood highlighting specific challenges in food availability, which have also been noted in another study examining barriers toward adherence to a MedDiet [72].

Cost was also identified as a barrier, particularly for key dietary components of the MedDiet, including fish, seafood, olive oil, and nuts. Previous investigations have documented that Mediterranean-style diets may be more expensive than a typical Western-style dietary pattern [73] due to its emphasis on fresh and minimally processed foods [74]. However, other studies found a negative association between MedDiet adherence and food expense [75]. This heterogenous financial impact was reflected in our findings, as some participants reported increased expenditure, while others managed to maintain or even reduce their costs by reallocating budgets from items such as alcohol or dining out. This budget reallocation strategy aligns with earlier research suggesting that prioritising spending on fresh produce over discretionary items can effectively manage food costs [75,76], which is a strategy that is particularly relevant given the current trends in rising food costs, where healthy foods are becoming disproportionately more expensive than unhealthy options [77].

External influences, including unsupportive or misaligned eating practices among family members and colleagues, further complicated the implementation of the MedDiet intervention in this study. These findings are consistent with previous studies that have identified negative social influences as important barriers to dietary adherence [78,79,80]. Conversely, the presence of supportive social networks has been shown to facilitate MedDiet adherence [78] and is frequently highlighted as beneficial in interventions in both healthy eating [81,82] and MedDiet interventions [83]. The importance for women to feel supported in their PCOS management was reiterated in systematic reviews by Guan et al. [84] and McGowan et al. [23], who concluded that women wanted positive support and feedback from friends, peers, and healthcare providers.

A central facilitator identified in our study was the acquisition of knowledge. Participants benefited from tailored educational resources, practical meal-planning suggestions, and a clear understanding of MedDiet principles and their associated health benefits. This is in line with prior research emphasising the importance of nutrition knowledge in improving diet quality [85,86]. Given the unique dietary challenges faced by women with PCOS, interventions that empower individuals with actionable knowledge may be particularly beneficial. These findings align with previous research, indicating that higher nutrition knowledge scores are positively associated with MedDiet adherence, even in non-Mediterranean cohorts [87,88]. Moreover, the documented health benefits of the MedDiet were highlighted as a facilitator to adherence by participants in the present study. This reflects similar findings in populations, such as women of childbearing age and those managing chronic illnesses, whereby health knowledge has been a significant facilitator of dietary adherence [78,80]. Nonetheless, the translation of knowledge into sustained behavioural change necessitates the integration of practical education and training to overcome negative beliefs [89]. Thus, strategies to overcome identified barriers to facilitate greater adherence have been recommended for future dietary implementation. The necessity for enhanced training and education among nutrition professionals is also underscored by these findings. Previous research has shown that less than 50% of Australian dietitians recommend the MedDiet for chronic disease management (such as CVD and type 2 diabetes), a reluctance attributed in part to gaps in knowledge and training, and a lack of ready-made patient resources [90,91]. This reluctance is also not unique to the Australian healthcare setting; similar trends have been reported internationally, with studies showing that 35% of dietitians in the United Arab Emirates rarely or never recommend the MedDiet to patients [92]. These findings further highlight the need for improved education and resources to support dietitians in incorporating the MedDiet into chronic disease management. Addressing these gaps through targeted professional development is essential for promoting patient-centred care and the effective implementation of individualised dietary interventions.

Beyond the practical issues of capability and opportunity, our qualitative data highlight the critical role of motivational factors in dietary adherence. The MedDiet was particularly well-received when presented as a flexible, nutrient-inclusive, and weight-neutral approach. These are characteristics that resonate with the needs of women with PCOS. Participants reported that the diet’s flexibility allowed them to incorporate foods they enjoy without the psychological burden of calorie counting or rigid restrictions. This finding is consistent with previous studies that underscore the importance of personalised [93], enjoyable lifestyle advice to promote adherence [94,95] and contrasts with the difficulties encountered when adhering to overly restrictive diets that result in significant difficulties reconciling dietary commitments with personal, social, and professional responsibilities, ultimately compromising long-term adherence [54,96,97,98]. The weight-neutral approach adopted in the present study is of particular relevance given the high prevalence of weight-related stigma and the risk of disordered eating among women with PCOS [23,99]. Shifting the focus from weight loss to overall health improvements, such as enhanced insulin sensitivity and reduced CVD risk, may be a more sustainable and psychologically supportive strategy for women who prefer non-weight-focused approaches; however, this may not be the priority for everyone. This aligns with a recent systematic review indicating that while some women with PCOS value lifestyle interventions that prioritise holistic health over weight loss alone, others may prefer approaches that align more closely with their personal health goals [23].

Moreover, in-depth interviews conducted in the present study revealed that women experienced improved energy levels, mood, and overall well-being, which served as strong motivators for sustained adherence. This aligns with key facilitators of implementing the PCOS lifestyle management guidelines, as participants believed these changes would help alleviate PCOS symptoms, reduce associated health risks, and enhance overall well-being [54]. Existing evidence for Mediterranean diet adherence supports these findings, with studies linking adherence to lower depression risk [100] and improved health-related quality of life scores [101]. Additionally, Jack et al. [102] reported that participants randomised to the Mediterranean diet intervention of an RCT experienced significant reductions in self-reported depression and anxiety symptoms after 12 weeks. Given the lower quality of life scores [103] and increased risk of mood disorders [104] among women with PCOS, further research is warranted to examine the direct impact of MedDiet adherence on psychological well-being in this population.

The acceptability of a MedDiet intervention has also been investigated in other non-Mediterranean countries, albeit in non PCOS populations. For example, in a sample of *n* = 67 middle-aged Northern Europeans at high risk for CVD, Moore et al. [105] reported that the barriers toward adopting a MedDiet were consistent to those of general healthy eating principles as well as those reported in the present study, including perceived expense, time commitment, access and availability, limited knowledge, lack of cooking skills, and a resistance to changing established dietary habits and practices. Similar barriers have also been reported in the United States, Netherlands, United Kingdom, and Ireland [106]. However, del Campo et al. [107] explored both the feasibility and acceptability of a culturally adapted MedDiet intervention to reduce CVD risk in low-income Hispanic American women, where the authors reported high intervention engagement and acceptability and an improvement in self-reported dietary behaviours. In recent years, the transferability and acceptability of a MedDiet in non-Mediterranean countries has garnered global interest, as evidenced by efforts to translate and culturally adapt MedDiet adherence tools in countries including China [108], Canada [109], and Brazil [110,111,112]. Additionally, key dietary principles of the MedDiet have also been successfully integrated into the American dietary guidelines, where they are recognised as an alternative to traditional recommendations. Cultural adaptation of a dietary pattern, such as the MedDiet, may promote longer-term adherence at a population level; however, in order to attain the proposed health benefits associated with MedDiet adherence, preserving key dietary elements (such as olive oil, nuts, legumes, and a low intake of red and processed meats) will be important [113]. Moreover, despite the potential therapeutic benefits of a MedDiet for the clinical management of PCOS [44], it is possible that alternate plant-based dietary patterns (e.g., Asian, Dietary Approaches to Stop Hypertension (DASH)) may also benefit PCOS management. However, the evidence for PCOS management is limited.

### Limitations and Strengths

Several limitations of this study should be acknowledged. The predominantly Australian-born sample limits the generalizability of these findings to a more ethnically diverse group of women. Additionally, the volunteer nature of recruitment suggests that participants may have had a higher motivation at baseline than the broader population of women with PCOS, potentially biassing the results [114]. The dual role of both delivering the intervention and conducting the interviews may have also introduced social desirability bias [115], although efforts such as memoing and researcher reflexivity were employed to mitigate this risk [116]. Moreover, while the survey was informed by the COM-B model, additional exploration of motivational components during data integration may have further enriched the findings.

Nevertheless, this study exhibits several notable strengths. It is among the first to explore the experiences of women with PCOS undergoing a MedDiet intervention, thereby providing unique insights into the behavioural determinants of dietary adherence. The integration of the COM-B model and the TDF in both the intervention design and data analysis provided a robust theoretical foundation, enabling a comprehensive examination of the cognitive, emotional, social, and environmental influences on adherence [56]. The mixed-method design further enhanced the depth of the analysis by integrating quantitative data with qualitative insights, and the application of the information power model suggests that the sample was adequately powered [66,117].

## 5. Future Directions and Conclusions

While barriers related to capability and opportunity continue to challenge adherence to a MedDiet among women with PCOS, the inherent motivational enablers of flexibility, weight neutrality, and nutrient inclusivity underscore the potential of the MedDiet as a promising dietary strategy. Future interventions should incorporate behavioural strategies that address these practical barriers and enhance motivational factors. In addition, there are potentially longer-term issues with transferability and acceptability of a MedDiet in non-Mediterranean countries, specifically maintaining core dietary components which have a known health benefit but also culturally adapting the MedDiet pattern to promote wider population-level uptake without widening socio-economic inequalities. Such challenges should be considered from an implementation science perspective. Moreover, targeted training and ongoing professional development for healthcare professionals in delivering weight-neutral, patient-centred nutritional advice is imperative for creating a supportive environment that facilitates sustainable dietary change. Lastly, this work offers critical insights into participants’ experiences with this intervention and is rigorously positioned to inform and strengthen a future feasibility study.

## Figures and Tables

**Figure 1 nutrients-17-01105-f001:**
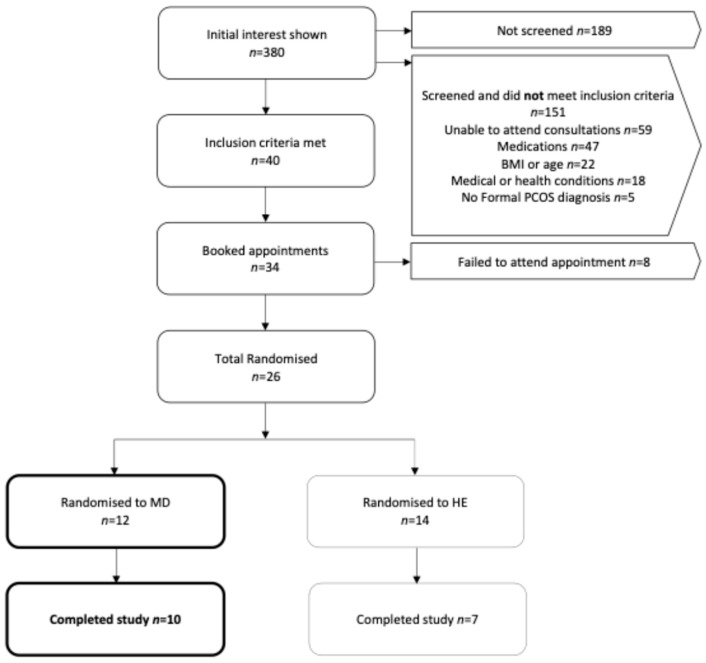
Flow diagram of participant recruitment.

**Figure 2 nutrients-17-01105-f002:**
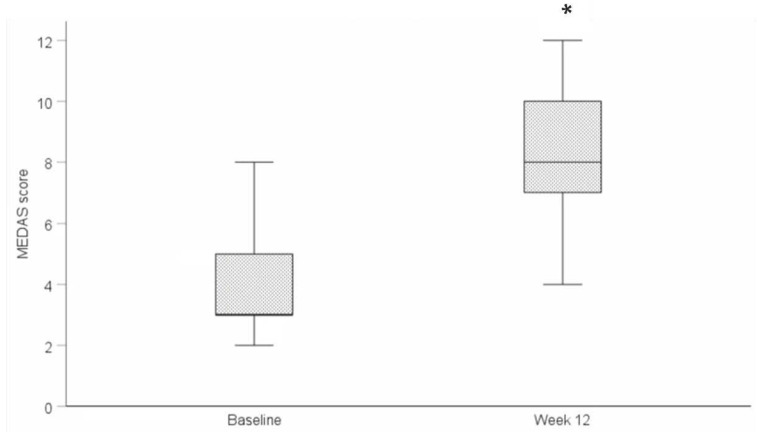
Mean change in Mediterranean Diet adherence scores (MEDAS) for the MedDiet group from baseline to week 12. * Statistical significance *p* ≤ 0.05.

**Figure 3 nutrients-17-01105-f003:**
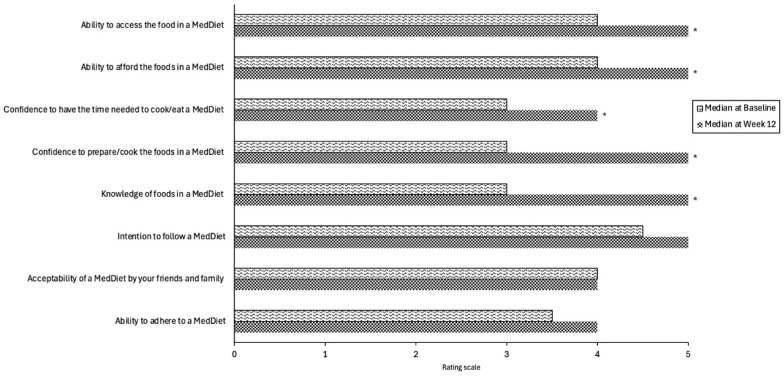
Participants’ self-reported median ratings of confidence and ability toward COM-B elements affecting MedDiet adherence from baseline to week 12. * Statistical significance *p* ≤ 0.05. Rating scale 1 none, 2 low, 3 neutral, 4 high, and 5 very high.

**Figure 4 nutrients-17-01105-f004:**
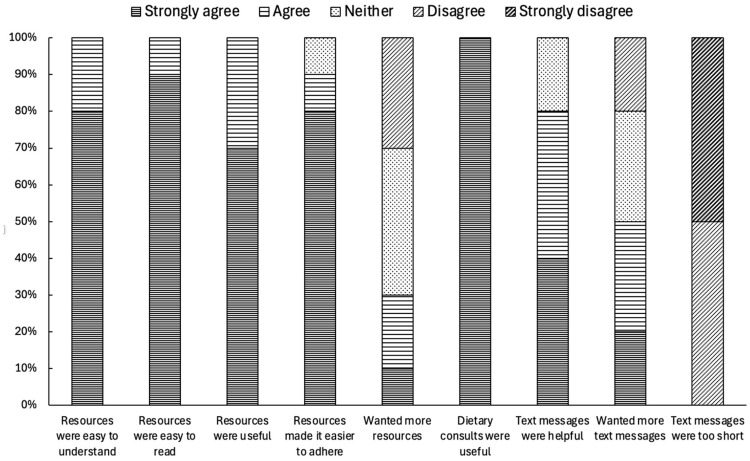
Participant assessment of intervention resources and delivery.

**Figure 5 nutrients-17-01105-f005:**
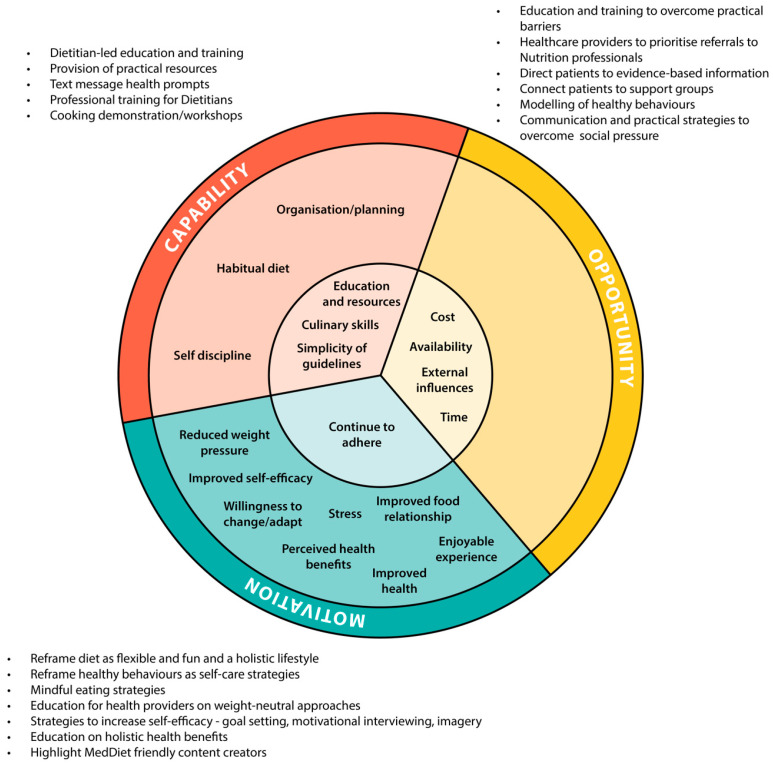
Integration of qualitative themes and quantitative data analysis, with a summary of recommended strategies for dietary implementation, mapped to the COM-B model. The inner circle represents areas where qualitative and quantitative findings align, while the surrounding sections depict themes derived solely from qualitative data.

**Table 1 nutrients-17-01105-t001:** MedDiet dietary protocol based on a traditional Mediterranean dietary pattern.

Include Daily	Include Weekly	Limit/Exclude
Four–six servings of wholegrains	Three servings of legumes	Red and processed meat
Five–six servings of vegetables	Two–three servings of fish and seafood	Discretionary foods
Two–three servings of fruit	One–three servings (100–150 g) poultry	Butter, cream, vegetable oils
One–four tablespoons extra virgin olive oil	Three servings (200 g) Greek yoghurt	
Swap to maximum of 200 mL red wine with main meal (only if alcohol is normally consumed)	Four–six eggs	
	Three servings nuts	

**Table 2 nutrients-17-01105-t002:** Baseline characteristics of participants.

Characteristic	MedDiet (*n* = 12)
	Mean ± SD
Age (years)	30.0 ± 5.5
MEDAS	4.4 ± 2.0
Weight (kg)	101.7 ± 18.8
BMI (kg/m^2^)	37.1 ± 7.0
Highest level of education	No. (%)
Year 10	0 (0)
Year 12	1 (8)
Trade	3 (25)
Bachelor	8 (67)
Country of birth	
Australia	9 (75)
Brazil	1 (8)
India	1 (8)
Argentina	1 (8)
New Zealand	0 (0)
Household income (AU$)	
25,000–74,999	3 (25)
75,000–124,999	7 (58)
125,000–174,999	1 (8)
175,000+	1 (8)
Smoking status	
Current smoker	1 (8)
Never smoked	8 (67)
Former smoker	3 (25)
Other health conditions	
Yes	6 (50)
No	6 (50)
Taking medications	
Yes	2 (17)
No	10 (83)
Taking supplements	
Yes	4 (33)
No	8 (67)

Abbreviations: BMI, body mass index. MEDAS, Mediterranean Diet Adherence Screener.

**Table 3 nutrients-17-01105-t003:** Influencers of adherence to the Mediterranean diet intervention, mapped to the COM-B and TDF with representative quotes.

COM-B	TDF	Theme from Qualitative Analysis	Quote
**Capability** Psychological	Knowledge	Education and resources (F)	Definitely the resources and all the information that you’ve provided every week, the initial pack was really good or where to go to find information and the Food pyramid there food Mediterranean food pyramid that helps with servings as well. P23
	Memory, attention and decision processes	Simplicity of guidelines (F)	It actually wasn’t that hard to follow. It was actually quite easy once you got into it. It actually was a fairly simple process. P27
	Behaviour regulation	Habitual diet (B)	I think the little bit I struggled with was red meat, because in my diet I have more of red meat throughout the week, and less of chicken and fish. P30
		Organisation/planning (B, F)	So the weeks where I was like, alright, cool, I’m gonna sit down and I’m gonna write a plan like what I’m gonna eat for every day and then I didn’t do it then I haven’t done any kind of meal prep or I haven’t got anything ready there waiting…that’s when you kind of get stuck and like, alright, what am I going to eat? P01 Being organised, yes, I’m planning like breakfast and snacks in advance for myself. That really helped. P17
**Capability** Physical	Skills	Culinary skills (B, F)	…like cause with fish and things I wasn’t really like a good cook at fish. I had to like research like different ways of cooking it. P26 And I suppose like feeling quite confident with my cooking skills. That certainly helped as well. P17
**Opportunity** Social	Social Influences	External influences (B, F)	So like partners that are fussy eaters or like work colleagues that like eat chocolate or have like, always having morning teas or afternoon teas. P26 …for the most part with with my direct family, everybody was always really understanding and and really welcoming of it as well, yeah. If anything, it probably brought me closer to my European grandparents, cause I was like asking them all about how they cook and any recipes that they had that I could sort of take on. P17
**Opportunity** Physical	Environmental context and resources	Time (B, F)	You can’t just get something and heat up or have something that’s in a package that you don’t have to prepare so that can sure add some challenges sometimes from a time point of view yeah. P9 But it’s an easy diet to follow, I think it doesn’t take much time. P30
		Cost (B, F)	And then the cost of that as well, yeah, you know, we were at the fresh produce market, and we did get some seafood while we were there but it was really expensive. P17 I don’t necessarily think it’s more expensive than like any other diet. So, I don’t really think that’s a factor. P13
		Availability (B, F)	I guess just I wish there was also more availability in terms of places around like there there are, but I just feel like when I was going to functions and things Like that for work that there wasn’t actually anything that like there was not as many things that I would have hoped that could have been incorporated into it. P6 the ingredients also a very go to ingredients. It’s nothing, fancy or out of the box ingredients. So, I didn’t have to go and specially source anything specifically. P30
**Motivation** Reflective	Belief about capabilities	Self-discipline (F)	I would say some self-discipline as well, especially when it came to you know if I wanted to like go out…it was very much like you see everyone else ordering all these lovely things and I’m like oh probably shouldn’t you know, so definitely self-discipline. P6
		Improved self-efficacy (F)	There are things that I want to make better and take that kind of next step of following it, and now maybe start to incorporate. Well, let’s maybe incorporate some like exercise on top of that [MedDiet] and make sure that I’m being really specific about. P23
	Optimism	Improved food relationship (F)	I did feel like it It actually felt nice to like approach a way of eating that wasn’t just centred around having a certain amount of calories like it did, just kind of feel a bit more liberating in a way. It was like you just eat really, and get these things in your diet, and that I just felt was a bit more freeing. P09
	Intention	Willingness to change/adapt (F)	So yeah, I guess I would say just willingness to change and learn and adjust as needed. P13
		Continue to adhere (F)	Oh, good. I’m gonna keep doing it. P31
	Belief about consequences	Perceived health benefits (F)	Helped my insulin resistance hopefully and then I know after this 12 week, I’m obviously hoping to continue it as well, so maybe help with my fertility as well, which would be great. P17
**Motivation** Automatic	Reinforcement	Improved health (F)	Yeah, so had more energy, quality of sleep improved for sure. P31
	Emotion	Stress (B)	I think just work stress honestly, just stress because when I come home like sometimes you just had such an average day that you’re just wanting to just grab whatever. P06
		Reduced weight pressure (F)	My biggest factor would probably be not having that weight loss pressure. Yeah, to be honest, because I’ve done all kinds of diets and the majority of them haven’t worked out and a lot of it, even if I have been following it, a lot of it was I was so stressed and put so much pressure on myself to lose weight or whatever not. And it just did not work. P01
		Enjoyable experience (F)	It’s just been fun. I’ve enjoyed doing it so, and and as I said, it’s the only thing I’ve ever been able to stick to past, you know, a week. P31

Abbreviations: F, facilitator B, barrier.

## Data Availability

The data generated during and/or analysed in this study are available from the corresponding author upon reasonable request due to privacy.

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
