# Peer review of "A Pilot Study on Ad Libitum Mediterranean Diet Intervention for Women with PCOS: A Mixed-Methods Exploration of Acceptability, Adherence, and Participant Lived Experience"

_nutrients, 2025, doi:10.3390/nu17071105_

Round 1

Reviewer 1 Report

Comments and Suggestions for Authors

An ad libitum Mediterranean Diet intervention for Women with PCOS: a mixed methods exploration of acceptability, adherence and participant lived experience.

The study is interesting, and you can find my appraisal as follows.

Introduction: In the introduction, you need to characterize obesity in PCOS in a better way since it is one of the conditions that can affect the quality of life in these patients. Moreover, you need to consider that in the Mediterranean area (including both the south of Europe and the north of Africa- Middle East), PCOS shows a high prevalence among women of fertile age.  This can be found in population studies that assessed the presence of polymorphisms associated with PCOS. I am referring to a study from Amin et al. (not mine). Moreover, it is plausible that women in these areas are very familiar with the Mediterranean diet, despite the fact the increase in BMI of populations (i.e. Italy) where this diet was usually present. According to me, this should be taken into consideration, here or in the following sections. Similarly, despite the healthy foods in the Mediterranean Diet and their quality, people cannot control the quantity of the food, especially those that contain carbohydrates, but also fish,  or EVO. It is only a suggestion. The hypotheses in line 120 need to be stated in a better and clearer way since I suppose that you have a control group (non-PCOS) BMI >25, or PCOS with BMI >25 divided in MedDiet and Control. So, you need to explain this better (detailed explained in the methods).  

Methods: The first sentence of the section is not clear. Please, remove or fix it. The survey should be added as an appendix. However, the methods are described in a rigorous way.

Results: The results are interesting and reported in a good way. I am curious about the anthropometric and metabolic results, but I suppose that will be published in a different paper. If it is, please add a brief statement.

Discussion: The section is interesting and well-written. However, I have found it a bit Australian-related, and this should be extended to other countries that usually do not have a MedDiet, such as western high-income countries. Moreover, despite the fact that the Mediterranean diet has a positive and preventive effect on CVD, other diets from other countries (such as Asian, such as Japan) could have a similar impact. Moreover, lines 615-620 contain considerations that could be more appropriate in the Conclusion. The future directions need to be improved.

Reviewer 2 Report

Comments and Suggestions for Authors

The work submitted by Mantzioris et al., titled: "An ad libitum Mediterranean Diet intervention for Women with PCOS: a mixed methods exploration of acceptability, adherence and participant lived experience" is an interesting mixed methods human study aiming to investigate the association between Mediterranean Diet in women with PCOS in terms of acceptability, adherence and experience. 

This is an interesting approach presented in a well designed and crafted manuscript. The work has significant potential for clinical implications, and touches upon an important and most practical aspect of dietary interventions and counseling that of acceptance and adherence to a dietary regime.

The work is well presented and the read flows nicely. The reviewer would like to offer a few points below for the authors' consideration aiming at strengthening further an already strong paper.

  1. BMI does not have units.
  2. Consider providing a rationale for the sample size determination.
  3. Consider discussing to what extent the questionnaires used are validated for the specific population.
  4. Consider discussing further the different approaches to Mediterranean Diet based on the ethnic background differences of participants.
  5. Understandably the n of the study is small. Thus, it may be more accurate to term it as a pilot study.

Good job overall 

Round 2

Reviewer 1 Report

Comments and Suggestions for Authors

The authors have addressed all my concerns. The manuscript is, according to me, suitable for publication.